

# Blue hypertext is a good design decision: no perceptual disadvantage in reading and successful highlighting of relevant information

Benjamin Gagl

Department of Psychology, Johann Wolfgang Goethe Universität Frankfurt am Main, Frankfurt am Main, Germany
Center for Individual Development and Adaptive Education of Children at Risk (IDeA), Frankfurt am Main, Germany

Corresponding author
Benjamin Gagl,
gagl@psych.uni-frankfurt.de

## ABSTRACT

**Background**. Highlighted text in the Internet (i.e., hypertext) is predominantly blue and underlined. The perceptibility of these hypertext characteristics was heavily questioned by applied research and empirical tests resulted in inconclusive results. The ability to recognize blue text in foveal and parafoveal vision was identified as potentially constrained by the low number of foveally centered blue light sensitive retinal cells. The present study investigates if foveal and parafoveal perceptibility of blue hypertext is reduced in comparison to normal black text during reading.

**Methods**. A silent-sentence reading study with simultaneous eye movement recordings and the invisible boundary paradigm, which allows the investigation of foveal and parafoveal perceptibility, separately, was realized (comparing fixation times after degraded vs. un-degraded parafoveal previews). Target words in sentences were presented in either black or blue and either underlined or normal.

**Results**. No effect of color and underlining, but a preview benefit could be detected for first pass reading measures. Fixation time measures that included re-reading, e.g., total viewing times, showed, in addition to a preview effect, a reduced fixation time for not highlighted (black not underlined) in contrast to highlighted target words (either blue or underlined or both).

**Discussion**. The present pattern reflects no detectable perceptual disadvantage of hyperlink stimuli but increased attraction of attention resources, after first pass reading, through highlighting. Blue or underlined text allows readers to easily perceive hypertext and at the same time readers re-visited highlighted words longer. On the basis of the present evidence, blue hypertext can be safely recommended to web designers for future use.

## INTRODUCTION

The Internet plays an important role in our daily life. One of the first but also most critical advantages of the Internet is the use of hypertext. Hypertext allows the web designer to efficiently link important snips of text to additional information. Thus, hypertext works by

replacing referencing in printed text and eliminating effortful literature searches. The most common implementation of hypertext, embedded as blue underlined text, was prominently criticized (*Nielsen, 1999*). It was argued that choosing blue as text color is a poor choice as only about 2% of retinal cells are sensitive to light with short wave-lengths, which give rise to the perception of blue. As a consequence, blue hypertext might reduce reading speed due to hampered foveal processing. This would be unfortunate since it would limit the general increase of effectiveness of text processing introduced by hypertext. In addition, color sensitive retinal cells are most prominent in the fovea of the eye and their number decreases massively towards para- and extra-foveal regions of the retina. This massive reduction of color sensitive cells towards the para- and extra-foveal regions might also decrease the possibility to extract relevant parafoveal information from colored text in reading. In general, parafoveal preprocessing typically increases reading speed (for a review see *Schotter, Angele & Rayner, 2012*). Therefore, a decrease in reading speed resulting from both reduced parafoveal preprocessing and foveal perception would be drastic when considering how often hypertext is perceived. Such a decrease in reading efficiency would indicate that the use of blue underlined hypertext could not be recommended.

Recently, *Fitzsimmons, Weal & Drieghe (2013)* investigated the influence of colored words on eye movement measures and found a reduced skipping probability (i.e., the probability of not fixating a word) of blue words, which were embedded in single line sentences. Their paradigm allows to examine both foveal and parafoveal processing combined during silent reading of sentences. They found a reduced reading speed, in contrast to black text, for words written in gray but not for words written in other colors (e.g., blue). This finding indicates that contrast (black vs. gray) but not color (e.g., black vs. blue) hampers reading speed. For words presented in saturated colors (e.g., blue) they found a reduced skipping probability in contrast to black-presented words. In their second experiment, *Fitzsimmons, Weal & Drieghe (2013)* did not find reduced skipping for blue colored words. In this experiment, the blue colored words were embedded in a paragraph, which was presented in a realistic online context (i.e., Wikipedia page). No effect of color on skipping was found but an interaction of a word frequency manipulation (seldom vs. common words) and the text color could be detected in late eye-movement measures of go-past-time (i.e., the sum of all fixations after the target word was fixated first and before the next word in the sentence is fixated; this includes fixations on words previous to the target word) and total viewing time (i.e., all fixation durations on the target word).

The finding in the first experiment from *Fitzsimmons, Weal & Drieghe (2013)* of a reduced skipping rate for colored words can be interpreted in two ways: Either bottom-up perceptual processes are hampered due to a reduced parafoveal perceptibility of blue words, increasing the fixation probability. The reduced visual information extraction would indicate that the lack of parafoveal blue light sensitive retinal cells results in less reliable visual information extraction. This then would reduce, first, information extraction from parafoveal words and, second, reading speed as a consequence of hampered word recognition processes. The second way to interpret the finding of a decreased skipping probability for blue words is that top-down processes, which increase the fixation probability, reflect the learned association of hypertext to informative content.

Therefore, highlighted words might attract additional attentional resources in contrast to un-highlighted words. The finding of the second experiment from *Fitzsimmons, Weal & Drieghe (2013)* is in line with this highlighting hypothesis. There they found no skipping effect difference between blue and black written text but showed that late eye movement measures were prolonged in case seldom, blue written words were fixated. Furthermore, this finding might indicate that such a top-down process would need time to evolve. This is consistent with previous findings showing that late eye-movement measures (e.g., total viewing time) are increased in case top-down processes like context demands or effortful reading instructions were manipulated (*Radach, Huestegge & Reilly, 2008*).

To differentiate between these hypotheses the present study realized an invisible boundary paradigm (*Rayner, 1975*; for a revised version of the paradigm see *Gagl et al., 2014*). This paradigm allows researchers to estimate the parafoveal preview benefit by contrasting fixation times after perfect previews (no manipulation) in contrast to degraded previews, which limit parafoveal preprocessing benefits. The task of the participant is to read sentences silently as if they were reading a book or newspaper (i.e., as natural as possible). An invisible boundary is placed before a target word (see Fig. 1A). When the invisible boundary is crossed by an eye movement (i.e., saccade) the change from a degraded to an un-degraded target word presentation is realized during the eye movement. The increase of reading speed after the parafoveal presentation of a normal word compared to the condition with a degraded word is interpreted as the parafoveal preview benefit. In our *Gagl et al., (2014)* study we were the first to use degraded parafoveal previews and showed such a degraded word before fixation elicited only less preprocessing benefit but no preprocessing costs. This is crucial as for parafoveal masks (e.g., different letter or X-masks), which were used originally instead of degradation, preview costs were described (see *Hutzler et al., 2013*; *Kliegl et al., 2013*). These costs potentially magnify parafoveal-preprocessing effects due to undesired processing of the mask. The boundary paradigm cannot be optimally implemented in case the skipping rate is expected to vary largely between conditions, as the estimation of the preview benefit relies on the fixation times on the target word. To realize high target word fixation rates, the predictability out of the sentence context was held low for the target words, which decreases skipping probabilities (*Fitzsimmons & Drieghe, 2013*; *Hawelka et al., 2015*). Therefore, low skipping rates, at the best-case floor effects, are expected to reduce the probability of finding differential effects in this measure. The effects of the present manipulations are expected in the fixation time measures of first fixation duration (i.e., the duration of the initial fixation), gaze duration (i.e., the summated fixation duration of all fixations during the first encounter), go-past time and total viewing time. In addition, two manipulations were implemented that allow to differentiate between the perceptual bottom-up (i.e., reduced reading speed for blue text) and the highlighting top-down hypothesis (i.e., more top-down attention is allocated to highlighted text). This was realized by manipulating color (blue vs. black) and underlining (underlined vs. not underlined) of the target words. The resulting design included the factors color, underlining and parafoveal degradation (see Fig. 1).

In case parafoveal bottom-up processing of blue stimuli is limited, a reduced parafoveal preview benefit in contrast to black words is expected. Limited foveal bottom-up processing

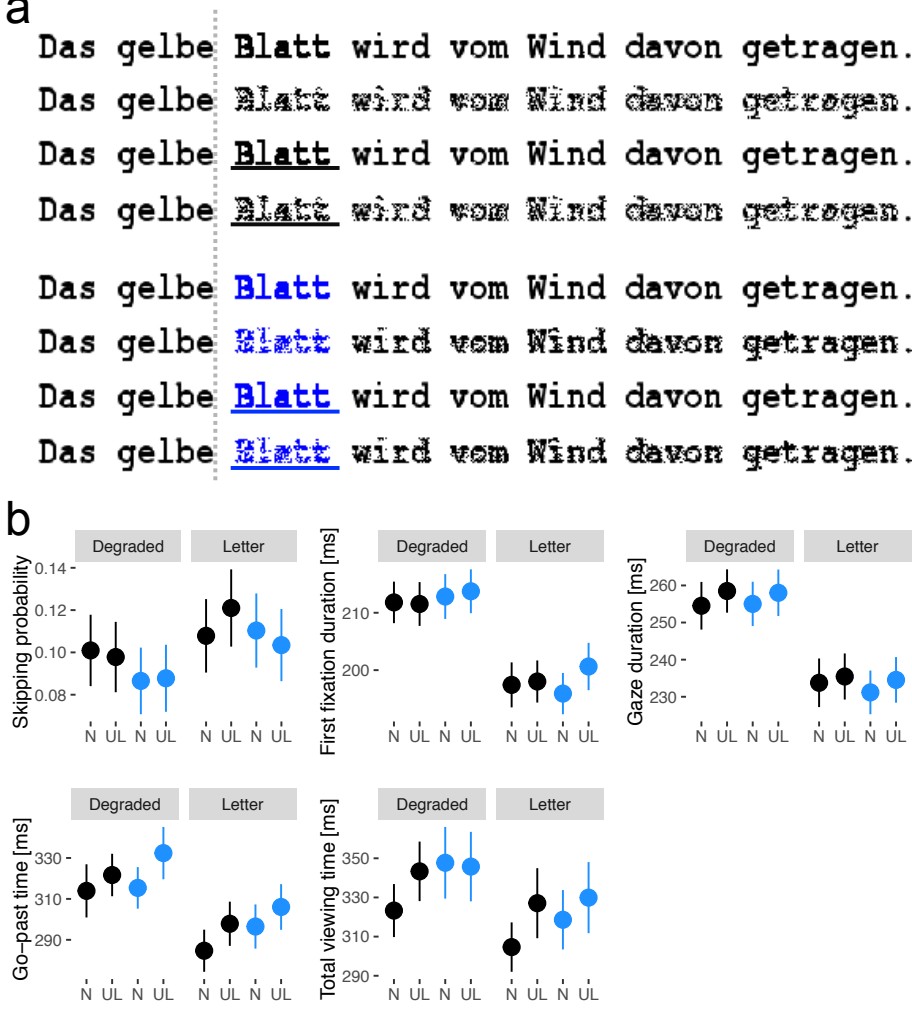

**Figure 1** **Stimulus presentation and eye movement data.** (A) Example sentence for all eight conditions (black not-underlined, black underlined, blue not-underlined, blue underlined in degraded and un-degraded versions) with the embedded target word *Blatt* (English: leaf). Before each target the gray line indicated the invisible boundary, which triggered the display change from degraded to un-degraded presentations in case a saccade crossed the boundary. The effect of parafoveal degradation can be visualized by introspection. Compare the percept of the target word after fixating either on the pre-target word of a degraded or an un-degraded target word. (B) Means and standard errors (vertical bars) of skipping probability, first fixation duration, gaze duration, go-past time, and total viewing time. Blue dots indicated blue words and black dots black words. *UL* indicates underlined presentation and *N* indicates normal presentation.

of blue text would result in higher fixation times of blue vs. black target words. This should be the case irrespective of underlining or parafoveal preview. The underlining aspect is crucial here as longer fixation durations for blue text in contrast to underlined black target words would exclude the possibility that such a difference would be the result of highlighting. Both parafoveal and foveal findings would indicate a hampered bottom-up processing of blue hypertext. Alternatively, if top-down processes that originate from highlighting influence the reading behavior, than the un-highlighted condition (i.e.,

black not-underlined targets) should receive less attention. In contrast, the highlighted words, blue not-underlined, blue underlined and black underlined targets, should receive additional attention reflected in longer fixation duration measures. If color or underlining results in independent highlighting effects (e.g., underlining effect when the target word is not colored), one can even recommend text underlining or coloring independently for hypertext use.

## METHODS

### Participants

Forty native German–speaking students (24 female; mean age: 23:2 years:month; standard deviation: 2:0) with normal reading speed measured by the unpublished adult version of the Salzburger-Lese-Screening (SLS; *Auer et al., 2004*; for the current state of the adult version see *Gagl, Hawelka & Hutzler, 2014* and normal or corrected-to-normal vision participated. One additional participant was excluded due to very slow reading (Percentile < 16). Participants gave informed consent and the research was approved by the ethics board of the University of Salzburg (EK-GZ: 20/2014).

### Apparatus

Movements of the right eye were recorded with a sampling rate of 2,000 Hz (EyeLink CL eye-tracker; SR-Research, Kanata, Ontario, Canada). Participants were seated about 52 cm in front of a CRT monitor (150-Hz refresh rate; screen resolution of 1024 × 768 pixels) and a forehead and chin rest stabilized their heads. The display change latency of the experimental setup was below 15 ms (for details see *Richlan et al., 2013*).

### Material

The manipulation of color and highlighting was realized with five letter target words embedded in sentences, which were matched on the most important word characteristics (e.g., orthographic similarity: OLD20 *Yarkoni, Balota & Yap, 2008*; word frequency: SUBTLEX database, *Brysbaert et al., 2011*; and predictability from sentence context, e.g., *Kliegl et al., 2004*). Furthermore, eight different versions of all sentences ($N = 320$) allowed the presentation of each sentence in one of the eight conditions (40 per condition; Fig. 1A). An equal number of participants ($n = 5$) were assigned to each of the eight versions of the sentences (Latin square design). The parafoveal preview manipulation was realized by randomly replacing 45% of the black or blue pixels of the presented letters (for details, see *Gagl et al., 2014*). This procedure distorted the parafoveal percept of the target words without inhibiting lexical processing. The sentences were presented in a mono-spaced font (single character width: 0.3° of visual angle) and target words were never at the first, second, or final position of the sentences (All stimuli are available at https://osf.io/8c57w/).

### Procedure

A 3-point calibration of the eye tracker preceded the experiment. Fixating between two vertical lines in the left margin of the monitor triggered sentence presentation in such a way that the participants' fixation was at the center of the sentence's first word. The students

read silently for comprehension. Fixating an 'x' in the lower right corner of the screen terminated the trial and removed the sentence from the screen. After the presentation of about 50 randomly selected sentences, a question mark appeared on the screen and the experimenter orally presented comprehension questions, which the participants almost always answered correctly ($M = 98\%$).

All words after the target word were visually degraded to minimize potential influences of these words (i.e., particularly of $n + 2$, with $n + 1$ being the target word; see *Kliegl, Risse & Laubrock, 2007* and Fig. 1A). After crossing the invisible boundary at the end of the pre-target word, the target word and the remainder of the sentence were presented un-degraded (see Fig. 1A). Ten practice trials preceded the experiment. Recalibration was conducted after the practice trials, after a break halfway through the experiment, and when the fixation control at the start of a trial failed.

## Data treatment and analyses

Skipping probabilities, first fixation durations, gaze durations, go-past times and total viewing time are reported. First fixation durations, gaze durations and total viewing times shorter than 80 ms were removed from the data (for each measure < 1% of the data). Data analysis was administered with linear mixed effect models (LMMs) for the *log*-transformed fixation timing measures and generalized linear mixed effect models (GLMMs) for the skipping probability (this analysis is best suited to estimate binary data: skipped vs. fixated) with the lme4-package (*Bates et al., 2014*) in R. G/LMMs are suited for analyzing unbalanced data (e.g., due to skipping of target words). Color, underlining, degradation and all interactions were included in the models as fixed effects (treatment contrast in relation to the baseline of black, not underlined, degraded words). Random effects were estimated for the intercepts of both participants and items. In addition, the random slopes for the fixed factors were added to the model until an additional parameter did not allow the model to converge. In case adding another level to the random effect structure resulted in a not converging model, one of the other two factors was introduced into the model and the model was refitted. If two models with the same number of random slope estimates converged, an ANOVA was used to compare the model fits and allowed to decide which model estimated the data better. This procedure resulted in the additional estimation of the random slope of degradation on the random effect of participant for the skipping probability. For the first fixation duration, the random slopes of degradation and color were estimated for the random effect of participant. For the gaze duration, the random slopes of underlining, color and degradation on the random effect of participant and the random slope of degradation on the random effect of item were estimated. For the go-past times, the random slopes of underlining, color and degradation on the random effect of participant and the random slope of underlining on the random effect of item were estimated. For the total viewing time the random slopes of underlining, color and degradation on the random effect of participant were estimated. With this procedure, the most conservative converging models were selected (Data and analysis scripts are available at https://osf.io/8c57w/).

## RESULTS AND DISCUSSION

As expected skipping probability, presented in Fig. 1B, was not reliably affected by color, underlining or degradation (see Table 1). The present study reports low skipping probabilities between 8 and 12%, when compared to the Fitzsimmons study with skipping probabilities up to 27%. Contrasting the differences of the two studies in the word frequency of the target words might explain the overall difference in skipping probability as in the present study words with low frequency were used and Fitzsimmons used words with higher frequencies. The result of the present study indicates that fixation rates of the target words are comparable indicating a floor effect for cognitive influences on word skipping.

In contrast, eye movement measures based on fixation durations during first pass reading indicated a strong preview benefit but no effect of color or highlighting. This was shown by the reliably lower first fixation durations and gaze durations for un-degraded parafoveal presentation in contrast to degraded previews (see Fig. 1B). No reliable effects and interactions of color or underlining were found (see Table 1). This finding indicates that bottom-up perceptual processing preceding word recognition (i.e., in parafoveal vision) was only influenced by degraded parafoveal previews but not reliably by word color or underlining.

The total viewing times, including all re-fixation times after the first encounter (i.e., re-reading) and the go-past times (including all fixations up to the moment the next word in the sentence is fixated), showed, in addition to a reliable degradation effect, a reliable interaction of word color and underlining and a reliable effect of underlining, respectively. Figure 1B clearly shows the origin of the interaction in total viewing times and the underlining effect in go-past times. The latter was indicated by lower go-past times for not underlined words in contrast to underlined words. The interaction for the total viewing times was reflected by the reduced fixation times for un-highlighted black-presented words in contrast to all other conditions including blue underlined, blue not-underlined and black underlined words (confirmed by post-hoc analysis: underlining effect for black targets; estimate $= 0.046$; SE $= 0.020$; $t = 2.29$; no underlining effect for blue targets; estimate $= 0.002$; SE $= 0.016$; $t = 0.12$). Both the interaction in total viewing times and the underlining effect in go-past times indicate that highlighting influences processing after first pass reading. The comparison of the two measures showed that re-readings after the target word was passed (reflected by total viewing times) reduced the effect of underlining for blue written words. This indicates that highlighting either by color or underlining increases the total reading times reflecting the allocation of additional attentional resources to highlighted words after the target word was first passed. Therefore, the reduced skipping probability of blue target words, described by *Fitzsimmons, Weal & Drieghe (2013)* in their experiment one, might also reflect a highlighting effect for sentences in which target word skipping can be realized to a higher extent. Finally, the finding of their second experiment in which they observed longer go-past on blue low frequent words in contrast to black low frequent words could be replicated (color effect for not underlined words irrespective of preview; go-past times: estimate $= 0.029$; SE $= 0.013$; $t = 2.27$).

**Table 1** Intercepts and fixed effects of G/LMM analyses for skipping probability, first fixation duration, gaze duration, go-past times and total viewing time (all timing measures were log transformed).

| | Fixed effects | SE | |
|---|---|---|---|
| *Skipping probability* | | | Z-values |
| Intercept | −2.61 | 0.19 | 13.63 |
| Degradation (Deg) | 0.14 | 0.13 | 1.08 |
| Color (Col) | −0.18 | 0.13 | 1.48 |
| Underlined (Undl) | −0.04 | 0.12 | 0.32 |
| Deg X Col | 0.22 | 0.17 | 1.25 |
| Deg X Undl | 0.18 | 0.17 | 1.09 |
| Col X Undl | 0.05 | 0.18 | 0.30 |
| Deg X Col X Undl | −0.28 | 0.24 | 1.14 |
| *First fixation duration* | | | t-values |
| Intercept | 5.31 | 0.017 | 314.53 |
| Deg | **−0.072** | **0.012** | **6.12** |
| Col | 0.004 | 0.011 | 0.33 |
| Undl | −0.001 | 0.010 | 0.10 |
| Deg X Col | −0.009 | 0.014 | 0.65 |
| Deg X Undl | 0.004 | 0.014 | 0.24 |
| Col X Undl | 0.007 | 0.014 | 0.50 |
| Deg X Col X Undl | 0.011 | 0.020 | 0.54 |
| *Gaze duration* | | | |
| Intercept | 5.47 | 0.03 | 190.87 |
| Deg | **−0.098** | **0.015** | **6.73** |
| Col | 0.007 | 0.012 | 0.59 |
| Undl | 0.021 | 0.013 | 1.59 |
| Deg X Col | −0.013 | 0.017 | 0.79 |
| Deg X Undl | −0.015 | 0.017 | 0.91 |
| Col X Undl | −0.012 | 0.017 | 0.69 |
| Deg X Col X Undl | 0.021 | 0.024 | 0.89 |
| *Go-past time* | | | |
| Intercept | 5.62 | 0.04 | 151.07 |
| Deg | **−0.103** | **0.015** | **6.32** |
| Col | 0.026 | 0.016 | 1.69 |
| Undl | **0.037** | **0.015** | **2.26** |
| Deg X Col | 0.008 | 0.016 | 0.40 |
| Deg X Undl | −0.001 | 0.020 | 0.05 |
| Col X Undl | −0.004 | 0.020 | 0.20 |
| Deg X Col X Undl | 0.000 | 0.029 | 0.01 |
| *Total viewing time* | | | |
| Intercept | 5.63 | 0.04 | 127.03 |
| Deg | **−0.080** | **0.015** | **5.32** |
| Col | 0.031 | 0.017 | 1.77 |
| Undl | **0.046** | **0.018** | **2.59** |
| Deg X Col | −0.011 | 0.021 | 0.53 |
| Deg X Undl | −0.028 | 0.021 | 1.34 |
| Col X Undl | **−0.044** | **0.021** | **2.09** |
| Deg X Col X Undl | 0.040 | 0.030 | 1.33 |

**Notes.**
Significant fixed effects are highlighted in bold numerals.

In sum, the present study demonstrated that reading was not significantly hampered by blue text presentation. Thus, the current findings do not indicate a bottom-up perceptual disadvantage of blue underlined hypertext in foveal and parafoveal processing. In contrast, the increased total viewing time for highlighted stimuli indicates an additional allocation of attentional resources triggered by top-down processes. These processes might reflect the learned association of hypertext to informative snips of texts in the Internet. Using blue underlined stimuli effectively highlights hypertext without hindering parafoveal and foveal perceptual processes during reading. In addition, the present study shows that text underlining and color results in highlighting, which indicates that either could be used independently to optimally implement hypertext. In conclusion, blue underlined hypertext allows effective reading and, therefore, can be safely recommended to web designers for future use.

## ACKNOWLEDGEMENTS

I want to thank Arturo Hernandez for helpful discussions and comments on an earlier version of the manuscript. I also want to thank Susanne Eisenhauer, Kirsten Hilger and Edvard Heikel for comments on an earlier version and Agnes Altmanninger, Pia Schweitzer and Eva Daspelgruber for helping with the data acquisition. Finally, I want to thank Florian Hutzler for letting me use his laboratory and Stefan Hawelka since part of the current software used in the project was scripted in collaboration for previous studies.

### Funding

This research was supported by the Goethe University Frankfurt "Nachwuchswissenschafter im Fokus –Förderlinie A". The funders had no role in study design, data collection and analysis, decision to publish, or preparation of the manuscript.

### Grant Disclosures

The following grant information was disclosed by the author:
Goethe University Frankfurt.

### Competing Interests

The author declares there are no competing interests.

### Author Contributions

- Benjamin Gagl conceived and designed the experiments, performed the experiments, analyzed the data, contributed reagents/materials/analysis tools, wrote the paper, prepared figures and/or tables, reviewed drafts of the paper.

### Human Ethics

The following information was supplied relating to ethical approvals (i.e., approving body and any reference numbers):

Ethikkommission Universität Salzburg

Approval number: EK-GZ: 20/2014.

## Data Availability

Gagl, B. (2016, April 17). Reading Hypertext. Retrieved from https://osf.io/8c57w/.

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
