# Peer review of "Blue hypertext is a good design decision: no perceptual disadvantage in reading and successful highlighting of relevant information"

_PeerJ, doi:10.7717/peerj.2467_

## Round 0.1 · original submission · Minor Revisions

Please take all the reviews into account, including reviewer 1, who suggests rejecting the manuscript. Please indicate in your revision where all alterations are made according to each reviewer's feedback.

Reviewer 1 ·

Basic reporting

The manuscript is nicely written and it is easy to follow. Unfortunately, the foundation of the experiment is shaky. The two hypotheses need to be formulated in a clear way. One was based on an internet link (instead of literature on human vision) and the other was based on a conference paper (Fitzsimmons et al., 2013; the two experiments from the conference paper were not described in detail in the manuscript).

Experimental design

To have the whole picture, one needs to include some other controls as parafoveal previews, not only identity pairs.

Validity of the findings

The discussion of the data does not produce much advance in knowledge (there was some discussion on bottom-up and top-down processing but there was no theory behind the reasoning). For example, the final sentence: “In conclusion, the blue underlined hypertext implementation allows effective reading and, therefore, can be safely recommended to web designers for future use.” That’s not new, and this does not go beyond the experiments reported by Fitzsimmon et al. (2013).

·

Basic reporting

The paper meets the specified criteria.

Experimental design

The paper meets the specified criteria.

Validity of the findings

The paper meets the specified criteria.

Additional comments

51: "retinal cells are sensitive to blue."
This is inprecise language: "retinal cells are sensitive to light to short wave lengths; they give rise to the perception of blue" (or something similar)

General: Replace the following with less drastic adverbs (or delete them): "incredibly", "drastically"

·

Basic reporting

No Comments

Experimental design

No Comments

Validity of the findings

No Comments

Additional comments

This research builds on the finding that low contrast words require longer fixations in order to be processed due to the increased difficulty in reading them. Also, coloured words reduce skipping probability. There could be two possible reasons for this. The finding could be because of the low level factor of saliency, where the reader is attracted to salient words or it could be due to high level factors such as a learnt association that salient words are often hyperlinks in a Web environment. The aim of the current paper is to pull apart these two suggestions. In order to do this a boundary paradigm experiment was conducted.

The author used single sentences with a target word and they manipulated the preview (either degraded or actual), the colour of the target word (either blue or black) and whether the target word was underlined (underlined or not).

The authors find that degraded target words need additional processing due to their difficulty, but no perceptual disadvantage of blue or underlined stimuli with no additional processing needed to read them. But they do find an attraction of attentional resources after first-pass reading and the reader revisited the salient words. This suggests that using blue as the colour for hyperlinks is a good choice as it doesn’t hinder reading, but does highlight the importance of hyperlinked words.

This article is very interesting, relevant to the area and forms a good baseline for future research exploring the impact of hyperlinks on reading behaviour. I recommend it be published with only minor additions. It is a tight experiment with a solid hypothesis, experimental design and conclusion.

Comments:

The finding of reduced skipping compared to Fitzsimmons study could be due to the author using low frequency words whereas Fitzsimmons et al used higher frequency words.

For the analysis of the data some additional information is required to fit into the current convention for reporting LMMs.
1 - There is no intercept reported in the fixed effect estimates table.
2 - Was the LMM conducted using a treatment contrast with a baseline? OR was it something like successive differences contrasts where the intercept becomes the grand mean so the author can talk about main effects of each of the fixed effects? Which ever is used, the convention as of late is to include this information in the results when specifying the models. If a baseline of say black text, no underline, identical preview was used then just mention this in the text and mention it is a treatment contrast and that should clear any ambiguity.

·

Basic reporting

This is a nice report on a straightforward experiment with clear results, well-implemented and overall well-written. I have a few comments which I feel need to be addressed in a revision but I am sure none of them will be particularly difficult to deal with.

The authors reference our previous work on the influence of blue words on reading behaviour (Fitzsimmons, Weal & Drieghe, 2013). The discussion of this study should mention that we did one experiment outside of a website context (like the current experiment, just a blue coloured word embedded in a sentence) where we showed reduced skipping of the blue (indeed of all colours we tested) compared to the black word. However, in a second experiment where we did embed the words in a website context, we did not observe this reduced skipping of the blue word and we observed longer go-past times on a blue word when it was low frequent. The current study nicely replicates the second experiment with a lack of an effect on skipping and longer late reading times (total reading times in this experiment) on the blue word/hyperlink. Given the relevance, please mention this in the paper. I do not think it reduces the impact of the current findings, quite the contrary.

Experimental design

It is not clear from the description of the design how many sentences the participants were reading in total and how long the experiment lasted. At first I assumed it was a latin square design (participants read one of the possible 8 versions of each of the 40 sentences). However, later I read that after about 50 sentences, questions were asked. So either it was a latin square design and the authors need to mention how many fillers there were, or it was not and the participants read all 8 versions which would lead to big effects of re-reading the same materials, something which would need to be dealt with. Please clarify.

Validity of the findings

Please also examine the go-past times. When comparing go past times with total times, it can be established whether the longer times in late measures were due to re-reading initiated from the target word or later.

Additional comments

Minor comments:
* Background: “The present study investigates if foveal and parafoveal perceptibility of hypertest is reduced during reading.” Please specify that this concerns the hyperlinks/blue words compared to black/non-hyperlinked text.
* Line 96-98: “To investigate the highlighting hypothesis…, highlighting was manipulated separately by underlining.” The added value of underlining as opposed to just colouring is not completely clear to me for examining this hypothesis. Please elaborate.
* Line 131. Typo. “Brysbaert” instead of “Brysbeard”.
* Line 222. “For now I can only offer congratulations for those who were able to produce such as successful educated guess”. I would suggest dropping this line but can live with it if the author wants to keep it.

---

## Round 0.2 · accepted · Accept

Please contact me with any concerns prior to final submission.

·

Basic reporting

All comments were addressed and I believe the manuscript is complete in it's reporting of all effects and the discussion of those effects. The addition of go past times fully rounds off the analysis and I believe it is a well rounded manuscript for publication.

Experimental design

Experimental design was sound and requires no additional changes.

Validity of the findings

The findings and discussion of which are well covered. All analyses are now present and discussed and I have no further comments.

·

Basic reporting

No Comments

Experimental design

No Comments

Validity of the findings

No Comments

Additional comments

I found the author to be very responsive to the issues raised in the reviews and am happy to endorse this paper for publication!